# Concurrent Validity and Relative Reliability of the RunScribe™ System for the Assessment of Spatiotemporal Gait Parameters During Walking

**DOI:** 10.3390/s24237825

**Published:** 2024-12-07

**Authors:** Andrés Ráfales-Perucha, Elisa Bravo-Viñuales, Alejandro Molina-Molina, Antonio Cartón-Llorente, Silvia Cardiel-Sánchez, Luis E. Roche-Seruendo

**Affiliations:** Faculty of Health Sciences, Universidad San Jorge, Villanueva de Gállego, 50830 Zaragoz, Spain; arafales@usj.es (A.R.-P.); ebravov@usj.es (E.B.-V.); amolinam@usj.es (A.M.-M.); scardiels@usj.es (S.C.-S.); leroche@usj.es (L.E.R.-S.)

**Keywords:** biomechanics, gait analysis, inertial measurement unit, technology, wearable

## Abstract

The evaluation of gait biomechanics using portable inertial measurement units (IMUs) offers real-time feedback and has become a crucial tool for detecting gait disorders. However, many of these devices have not yet been fully validated. The aim of this study was to assess the concurrent validity and relative reliability of the RunScribe™ system for measuring spatiotemporal gait parameters during walking. A total of 460 participants (age: 36 ± 13 years; height: 173 ± 9 cm; body mass: 70 ± 13 kg) were asked to walk on a treadmill at 5 km·h^−1^. Spatiotemporal parameters of step frequency (SF), step length (SL), step time (ST), contact time (CT), swing time (SwT), stride time (StT), stride length (StL) and normalized stride length (StL%) were measured through RunScribe™ and OptoGait™ systems. Bland–Altman analysis indicated small systematic biases and random errors for all variables. Pearson correlation analysis showed strong correlations (0.70–0.94) between systems. The intraclass correlation coefficient supports these results, except for contact time (ICC = 0.64) and swing time (ICC = 0.34). The paired t-test showed small differences in SL, StL and StL% (≤0.25) and large in CT and SwT (1.2 and 2.2, respectively), with no differences for the rest of the variables. This study confirms the accuracy of the RunScribe™ system for assessing spatiotemporal parameters during walking, potentially reducing the barriers to continuous gait monitoring and early detection of gait issues.

## 1. Introduction

Walking is a low-demand physical activity that is essential for maintaining an optimal quality of life and well-being [1]. Furthermore, walking offers numerous advantages, including improved physical and mental health, as well as the prevention of chronic conditions such as heart disease, cancer, stroke and diabetes [2]. Of note, several previous studies have identified strong associations between altered gait patterns and neurodegenerative pathologies [3,4], risk of falls [5] and even mortality [6,7]. According to the Web of Science, over 24,000 studies have been published based on gait analysis in the last 10 years, which makes walking one of the most extensively researched physical activities.

In clinical practice, visual inspection is necessary to get a first impression of a person’s walking performance. However, this qualitative assessment is often subjective and therefore unreliable and inaccurate for biomechanical analysis. The literature suggests that a quantitative analysis should be performed to identify any pathology, provide information about disease progression and measure the efficacy of interventions [6]. In addition, clinical gait analysis plays a critical role in clinical decision making, particularly in the context of surgery. Research has demonstrated that incorporating gait analysis can refine proposed surgical procedures, lead to changes in final decisions and ultimately reduce the costs associated with surgery, highlighting its economic impact [8,9,10]. In this regard, the evaluation of gait biomechanics, using advanced technological devices, has become a crucial tool for diagnosing and detecting gait disorders. Motion capture gold standard methods, such as 3D motion capture, typically require multiple and expensive technical equipment and a large space to capture measurements. That is why there has been a move to produce low-cost and easy-to-use tools for assessing walking spatiotemporal variables, such as markerless motion capture or photoelectric cell systems, looking for a more natural environment [11,12]. In particular, the OptoGait™ system has gained popularity among sports scientists and clinicians for its ability to capture and analyze spatiotemporal gait parameters. The validity and reliability of this gait analysis system have been previously analyzed while walking [13,14,15,16,17] and running [18,19].

In a similar way, to reduce costs and make assessments easier, researchers have recently developed portable gait analysis equipment based on inertial measurement units (IMUs). These tools integrate a tri-axial accelerometer (to quantify linear accelerations), tri-axial gyroscopes (to measure angular velocities) and a tri-axial magnetometer (to determine heading direction). The combination of data obtained from these sensors allows precise assessment of segment orientations and joint angles. Therefore, inertial sensors provide a robust method for capturing diverse, objective and clinically relevant gait metrics, providing real-time feedback on spatiotemporal gait parameters and allowing researchers to measure participants in a more natural environment [11,20]. However, some of these devices have not yet been validated. The validity and reliability of a gait analysis system are essential for distinguishing whether observed changes in gait pattern are authentic or are simply the result of systematic measurement errors. In this sense, Kobsar et al. [21] conducted a systematic review and meta-analysis to evaluate the validity and reliability of various wearable inertial sensors (e.g., Opal, APDM Inc (Portland, OR, USA).; Physiolog, BioAGM (Lausanne, Switzerland.); IDEEA, MiniSun LLC (Fresno, CA, USA), among others) in assessing walking in healthy adults. Their findings demonstrated excellent validity and reliability of IMUs for measuring mean spatiotemporal parameters. However, the strength of these conclusions is constrained by the limited availability of high-quality studies for specific outcomes, with small and/or underpowered sample sizes (median sample size: 12; range: 2–95) representing a major limitation.

In this regard, the RunScribe™ system (Scribe Labs Inc., San Francisco, CA, USA), launched in 2015, is a portable device that is attached to the laces or heel of the shoe which provides valuable information about spatiotemporal gait parameters and many other motion metrics such as shock, braking or pronation. Interestingly, previous studies analyzed the validity and reliability of the RunScribe™ system while running [22] and the influence of the system placement on the accuracy of the spatiotemporal gait characteristics [23], whereas in walking, its validity has only been studied for kinetics (i.e., shock and braking forces) and kinematics (i.e., pronation excursion) [24]. However, an evaluation of the concurrent validity of this portable device on spatiotemporal gait parameters while walking is lacking.

Therefore, the aim of this study was to evaluate the concurrent validity and relative reliability of the RunScribe™ system for measuring spatiotemporal parameters during walking by comparing data with a widely used device for this purpose (i.e., the OptoGait™ system). This study will provide valuable insights into the real-time, real-world assessment of gait dysfunctions, demonstrating the potential of low-cost wearable devices as reliable and valid tools for gait analysis.

## 2. Materials and Methods

### 2.1. Participants

A sample of young and middle-aged adults between the ages of 20 and 50 with a low potential risk for significant gait abnormalities was recruited. Only individuals who voluntarily agreed to participate in the biomechanical gait study and met the inclusion criteria were ultimately evaluated. Those who presented orthopedic, vertigo, neurological, vascular, metabolic alterations or sequelae of an amputation that may alter or limit the locomotion were excluded from the study. All participants were advised to wear their usual footwear to enhance the ecological validity of the study and minimize potential gait alterations that could result from using shoes different from their normal choice for a long walk. Therefore, minimalist shoes were not permitted, and participants were encouraged to wear athletic or running shoes instead.

Data were collected in two biomechanical laboratories following the same procedure. Two experienced researchers with expertise in the use of RunScribe (Scribe Labs, Moss Beach, CA, USA) and OptoGait (Microgate, Bolzano, Italy) collaborated in the design and implementation of the experimental protocol. Data collection was later extended to a second laboratory. Both researchers strictly followed the calibration and setup protocols provided by the manufacturers to ensure consistency between the two biomechanical laboratories.

After the removal of duplicated and missing data, the database included three hundred and eighteen men and one hundred and forty-two women (age: 36 ± 13 years; height: 173 ± 9 cm; body mass: 70 ± 13 kg). The flow diagram (Figure 1) shows the data treatment. Informed consent forms were collected from all participants before the assessment after receiving detailed information about the study. This study complied with the ethical standards of the World Medical Association’s Declaration of Helsinki (2013). The participants were informed that they were free to leave the study at any time. The study was approved by the Ethics Committee of the corresponding institution (28/1/23-24).

### 2.2. Procedures

The testing protocol was designed to be carried out on one specific day. Participants were asked to walk on a motorized treadmill (WOODWAY Pro XL, Woodway, Inc., Waukesha, WI, USA) at 5 km·h^−1^, and the slope gradient was set at 0%. Participants wore comfortable clothing and their own footwear during the entire protocol. Before the gait analysis started, participants were allowed 10 min to adapt to treadmill walking, approaching the targeted speed [25]. Once participants felt stable and comfortable, a 60 s data capture period was recorded.

### 2.3. Materials and Testing

Body height (cm) and mass (kg) were measured using a precision stadiometer and weighing scale (SECA 222 and 634, respectively, SECA Corp., Hamburg, Germany) for descriptive purposes.

Spatiotemporal gait parameters were measured using two different devices, the OptoGait™ system and the RunScribe™ system. Both systems have the same temporal accuracy (±1 ms). To avoid potential influencing factors (i.e., asymmetry), the right leg was always the analyzed leg [26]. A total of 90 s of data was recorded simultaneously using both instruments. For analysis, the first and last 15 s of each capture were excluded, thus focusing on the central minute. The average values for each subject were included in the comparison between devices, with the data matched using the timestamp and subject code. Further information about the systems is described below:-The OptoGait™ system (Optogait; Microgate, Bolzano, Italy) consists of photoelectric cells placed along two 1 m-long bars. This system detects any interruption in the light connection between the bars which are placed on the lateral edges of the treadmill and 3 mm above the ground level. Spatiotemporal gait parameters were measured for every step during the treadmill test. The data were collected and analyzed by the OptoGait software (version 1.6.4.0, Microgate, Bolzano, Italy).-The RunScribe™ system (Scribe Lab Inc., San Francisco, CA, USA) consists of a 9-axis inertial measurement unit (IMU), 3-axis gyroscope, 3-axis accelerometer and 3-axis magnetometer, and was attached to the shoelace of the right leg. Once recorded, the data were downloaded from RunScribe’s website (https://dashboard.runscribe.com/runs accessed on 2 May 2024) into a csv file.

Finally, the data collected by the two devices were imported into Excel^®^ (v. 2016, Microsoft, Inc., Redmond, WA, USA).

The following spatiotemporal parameters while walking were considered (Figure 2):-Step frequency (SF): the number of steps per minute.-Step time (ST): time between the initial contact of one foot and the subsequent initial contact of the opposite foot, in seconds.-Step length (SL): distance between the initial contact of one foot and the subsequent initial contact of the opposite foot, in meters.-Stride length (StL): distance between two consecutive initial contacts of the same foot, in meters.-Stride length % (StL%): stride length normalized with height, in percentage; dividing the stride length by the height of the subject.-Stride time (StT): time between two consecutive initial contacts of the same foot, in seconds.-Contact time (CT): time from initial contact to toe-off, in seconds.-Swing time (SwT): time from the toe-off to the initial contact, in seconds.

### 2.4. Statistical Analysis

Descriptive statistics are represented as mean (±SD). Before data analysis, normal distribution and homogeneity were assessed. Normal distribution was determined by the Shapiro–Wilk test. Homogeneity was determined by Levene’s and White’s tests.

Bland–Altman plots and table (systematic bias and random error, limits of agreement as mean difference ±1.96 SD, percent of values within limits of agreement) [27] were constructed to assess the agreement between the measured and calculated values of gait variables for the OptoGait™ vs. RunScribe™ systems.

To compare data from the two devices, a t-test was conducted (by Welch’s test). The magnitude of the differences between values was also interpreted using Cohen’s d effect size (ES) (between-group differences). Effect sizes are reported as trivial (<0.19), small (0.2–0.49), medium (0.5–0.79) and large (≥0.8) [28].

To determine concurrent validity, a Pearson correlation analysis was performed between Optogait™ and RunScribe™ spatiotemporal parameters. The following criteria were adopted to interpret the magnitude of correlations between measurement variables: <0.1 (negligible), 0.1–0.39 (weak), 0.4–0.69 (moderate), 0.7–0.89 (strong) and 0.9–1 (very strong) [29]. Additionally, this analysis was complemented with a linear regression model.

Intraclass correlation coefficients (ICCs), as measures of relative reliability, were also calculated between Optogait™ and RunScribe™ for spatiotemporal values during walking. Based on the characteristics of this experimental design and following the guidelines reported by Koo and Li, a “two-way random-effects” model (ICC[2,k]), “mean of measurements” type and “absolute” definition for the ICC measurement were conducted. ICC values less than 0.5 are indicative of poor reliability, values between 0.5 and 0.75 indicate moderate reliability, values between 0.75 and 0.9 indicate good reliability, and values greater than 0.90 indicate excellent reliability [30].

In this study, the ICC(2,k) was chosen instead of the ICC(3,k) to assess the agreement between the instrument under validation and the gold standard. Although the ICC(3,k) is widely used in previous works because of its simplicity and its focus on specific agreement between fixed instruments, the ICC(2,k) allows capturing the variability inherent to the measurement process and generalizing the results beyond the instruments evaluated in this study. Although the values obtained with the ICC(2,k) may be lower due to its more stringent nature, this approach reinforces the validity of the new instrument by demonstrating its consistency under broader and more realistic conditions. The ICC(3,k) analysis is available in Appendix A.

The level of significance used was *p* < 0.05. Data analysis was performed using Python (version 3.10.10, Python Software Foundation License). Python is a free software environment for machine learning, data science and graphics.

## 3. Results

A description of the participants’ characteristics is shown in Table 1.

### 3.1. Agreement Between OptoGait™ and RunScribe™

Concerning the agreement between the two devices, Bland–Altman plots (Figure 3) show the systematic differences and random variability between the methods. The resulting graph is a scatterplot in which the Y-axis shows the differences between paired measurements (A-B), while the X-axis represents the average of these measurements ((A + B)/2). This approach helps to identify systematic bias (mean of the differences) and to calculate the limits of agreement (±1.96 standard deviations of the mean difference) [31]. The analysis revealed small systematic biases and random errors, with mean differences around 0, for all the variables: SF (0.07 ± 2), ST (0 ± 0.01), SL (0.01 ± 0.02), StL (0.02 ± 0.05), StL% (1.07 ± 2.83), StT (0 ± 0.02), CT (0.05 ± 0.02) and SwT (−0.05 ± 0.02). These small values suggest minimal discrepancies between the devices. Importantly, the Bland–Altman table (Table 2) indicates that over 90% of the values fall within the limits of agreement for all variables, reinforcing the practical agreement between the two methods. It should be noted that the definition of “small” differences is context dependent.

### 3.2. Comparative Analysis

Table 3 shows descriptive values and comparative analysis between systems. The *T*-test demonstrated no significant differences between the systems in SF (*p* = 0.849), ST (*p* = 0.28) and StT (*p* = 0.28), indicating strong consistency in these variables. Although significant differences were observed in SL, StL and StL%, Cohen’s d values for these variables indicate small effect sizes (SL: ES = 0.22; StL: ES = 0.22; StL%: ES = 0.25), suggesting that these differences, although statistically significant, are unlikely to be of practical relevance in this context. Cohen’s d values for CT (ES = 1.2) and SwT (ES = −2.2) indicate large effect sizes, highlighting meaningful differences between systems for these variables. However, Bland–Altman analysis reveals that the mean differences for CT and SwT are approximately 0, and the majority of data points fall within the limits of agreement. This indicates that, despite the large effect sizes, the differences between the systems are minimal in practical terms and likely not significant. These findings underscore the importance of integrating both statistical measures (e.g., effect sizes) and graphical methods (e.g., Bland–Altman) to provide a comprehensive assessment of agreement and to avoid overestimating the impact of statistical differences when their practical relevance is negligible.

To determine concurrent validity, a Pearson correlation analysis was conducted between the two devices. Strong correlations were obtained in SL (r = 0.79; *p* < 0.001), StL (r = 0.79; *p* < 0.001), StL% (r = 0.78; *p* < 0.001), CT (r = 0.82; *p* < 0.001) and SwT (r = 0.70; *p* < 0.001). Very strong correlations were obtained in SF (r = 0.93; *p* < 0.001), ST (r = 0.92; *p* < 0.001) and StT (r = 0.92; *p* < 0.001). Additionally, the linear regression model shown in Figure 4 described an almost proportional relationship between the two devices for all the variables. Finally, the ICC reported an excellent agreement in SF (ICC = 0.97), ST (ICC = 0.96) and StT (ICC = 0.96). Good reliability was observed in SL (ICC = 0.87), StL (ICC = 0.87) and StL% (ICC = 0.86). However, moderate agreement was observed in the CT (ICC = 0.64) and poor reliability in the SwT (ICC = 0.34).

## 4. Discussion

This study aimed to determine the relative reliability and evaluate the concurrent validity of the RunScribe™ system for measuring spatiotemporal variables during walking by comparing data with a previously validated device, the OptoGait™ system.

The main finding of this study was that the RunScribe™ system provided valid and reliable spatiotemporal data during walking for almost all variables, compared to the OptoGait™ system. An almost perfect agreement between the two devices was observed, with small systematic biases in all the variables. The percentage of values within the limits of agreement was over 92.5% for all variables, and the mean differences were negligible (≤ 0.5), except for the normalized stride length (1.07%). In addition, all spatiotemporal parameters reported a strong (0.70–0.82) or very strong (0.92–0.94) correlation between the systems. Although pairwise comparisons revealed significant differences between the two systems for some variables, the magnitude of these differences was small for SL, StL and StL% (0.21–0.25) and large for CT and SwT (1.21 and 2.19, respectively). Similarly, the RunScribe™ system revealed good or excellent reliability in terms of ICC for all variables, except for CT (ICC = 0.64) and SwT (ICC = 0.34).

In this regard, a previous systematic review, conducted in 2020 by Kobsar et al. [21], indicated that IMUs commonly used for the assessment of spatiotemporal parameters during walking demonstrated excellent validity and reliability (ranging from 0.81 to 0.94) for this purpose. In addition, it appears that many systems have difficulty in detecting toe-off events (i.e., BioStampRC, MC10 Inc.) [32] and show poor accuracy in reporting stance and swing times [32,33,34,35,36], which supports our findings. Nevertheless, the strength of the evidence of the studies included in the systematic review was questioned due to the small sample sizes of the included studies (median sample size was 12, range 2 to 95). In response to this limitation, the present study included 460 individuals, addressing the concerns raised by this systematic review.

In terms of sensor placement, other systems often detect gait events by positioning sensors in the lumbar region [37,38,39] or on the lower leg [39,40,41]. However, consistent with the findings of this study, the accuracy of gait events improves when sensors are positioned closer to the ground (i.e., APDM Opal system) [41,42].

Despite the overall good validity and reliability of RunScribe™ for walking gait analysis found in the current work, the results for SwT and CT slightly deviate from those reported in the aforementioned study [18]. Comparing IMUs and photoelectric cells, authors hypothesize that the observed differences in temporal variables may be partially explained by Lienhard et al. [15]. Specifically, the LEDs of the OptoGait™ system are positioned 3 mm above ground, causing heel contact to be detected earlier and toe lift-off to be detected later in the gait cycle. This leads to the recording of longer CT and shorter SwT. Similarly, Gindre et al. [43] assessed the reliability and validity of an accelerometer (Myotest) in comparison to a photocell-based system (OptoJump) for measuring running stride kinematics. Notwithstanding variations in the experimental design (i.e., running vs. walking), the authors found that the accelerometer recorded shorter contact times and longer flight times compared to the photoelectric system. In the same way, a recent study [19] assessed the validity and reliability of another wearable IMU (Stryd) against the OptoGait system, presenting comparable discrepancies between the two systems. Therefore, the data obtained in the current study agree with those reported by previous studies that compared accelerometers to photoelectric systems, further supporting the explanation provided by Lienhard et al. [15] for these observed differences.

As mentioned before, scientists have been exploring the potential of IMUs to perform gait analysis in a more natural setting and with less expensive and less time-consuming equipment. Particularly, García-Pinillos et al. [22] analyzed the spatiotemporal parameters of 49 runners and concluded that RunScribe™ seemed to be an accurate system for measuring spatiotemporal parameters during running on a treadmill at a comfortable speed. Conversely, no previous studies evaluated the concurrent validity and reliability of this wearable device on the accuracy of spatiotemporal parameters while walking. In this sense, current results seem to be consistent with previous evidence and highlight the applicability of the RunScribe™ system in a more practical approach.

Since methodological differences can be observed among the aforementioned studies, some points must be considered to properly interpret the comparisons. The first point that must be taken into consideration is the lack of studies that analyzed RunScribe™ while walking. To date, only one study [24] evaluated this system but focused on kinetic and kinematic variables such as shock, impact, braking forces and pronation excursion. Further, existing evidence on its validity and reliability was collected while running [22], which makes it not possible to properly compare the variables. Finally, the last point that must be considered is the reference system used in the study. While in the present study, a photoelectric cell system was selected as the gold standard, previous works compared spatiotemporal data obtained from RunScribe™ with a high-speed video-analysis system or 3D motion capture system [14,22,23]. However, the OptoGait™ system has been previously validated against these methods showing excellent accuracy for spatiotemporal parameters [44].

From a practical point of view and given the widespread use of both systems, it is essential for scientists and clinicians to recognize that both devices have demonstrated adequate reliability for walking assessment, and therefore, spatiotemporal parameters reported by these devices can be compared over time (when using the same device). However, it is equally important for users to understand the limitations of directly comparing data between these two devices, particularly in the assessment of contact and swing times. The results of the present study hold significant implications for clinicians and researchers utilizing IMUs to assess and investigate gait abnormalities. In addition, the large sample size included in this study makes its findings of great value and enhances its validity among larger populations of young adults. Future research should explore the applicability of these results to other populations, such as individuals with mobility impairments or different age groups, and investigate their appropriateness in overground conditions.

## 5. Conclusions

This study provides evidence for the validity and relative reliability of the RunScribe™ system for assessing several spatiotemporal parameters during walking. However, the findings indicate variability in the accuracy of temporal parameters, particularly contact and swing times. This limitation underscores the need for caution when interpreting these specific metrics and suggests the importance of further validation studies using gold-standard systems. The development of accurate wearable devices opens new possibilities for widespread use in healthcare contexts, potentially reducing the barriers to continuous gait monitoring and early detection of mobility issues. As a future line, it would be interesting to evaluate its accuracy during overground walking.

## Figures and Tables

**Figure 1 sensors-24-07825-f001:**
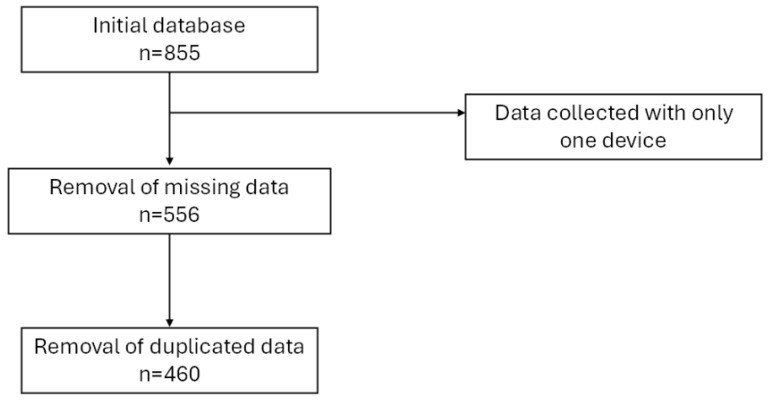
Flowchart of the database.

**Figure 2 sensors-24-07825-f002:**
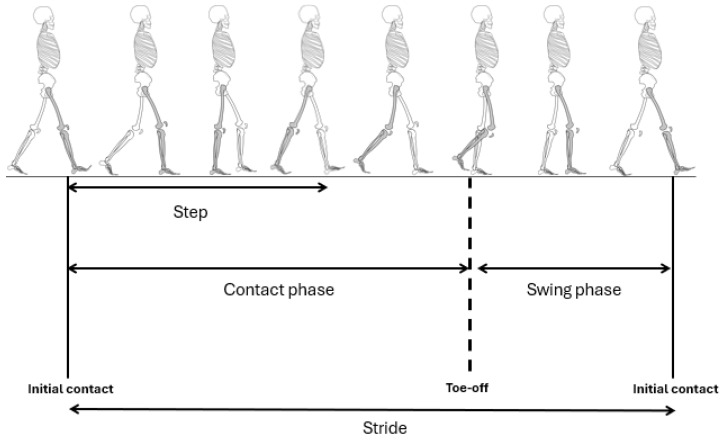
Gait cycle.

**Figure 3 sensors-24-07825-f003:**
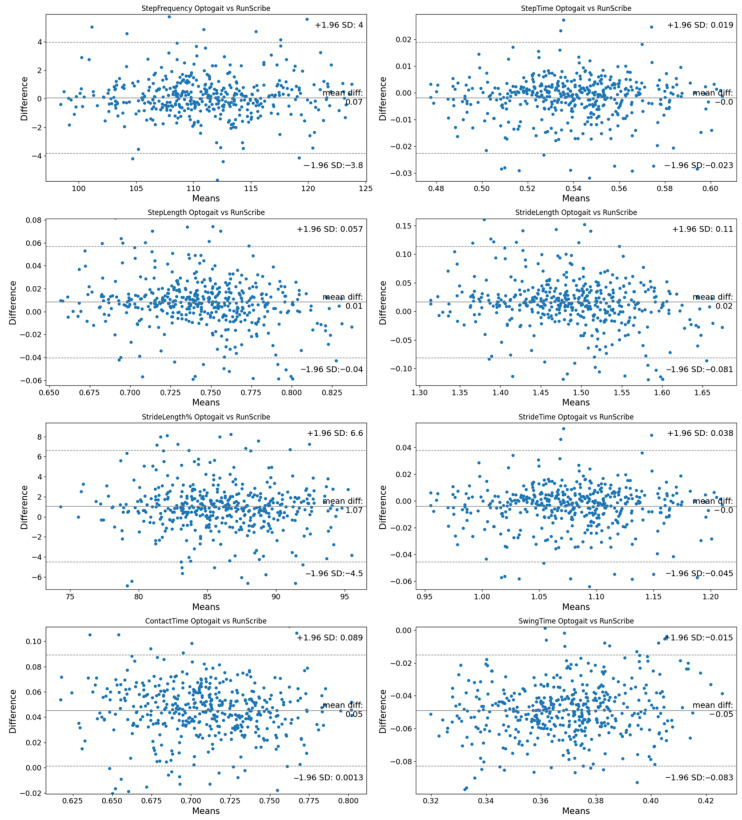
Bland–Altman plots.

**Figure 4 sensors-24-07825-f004:**
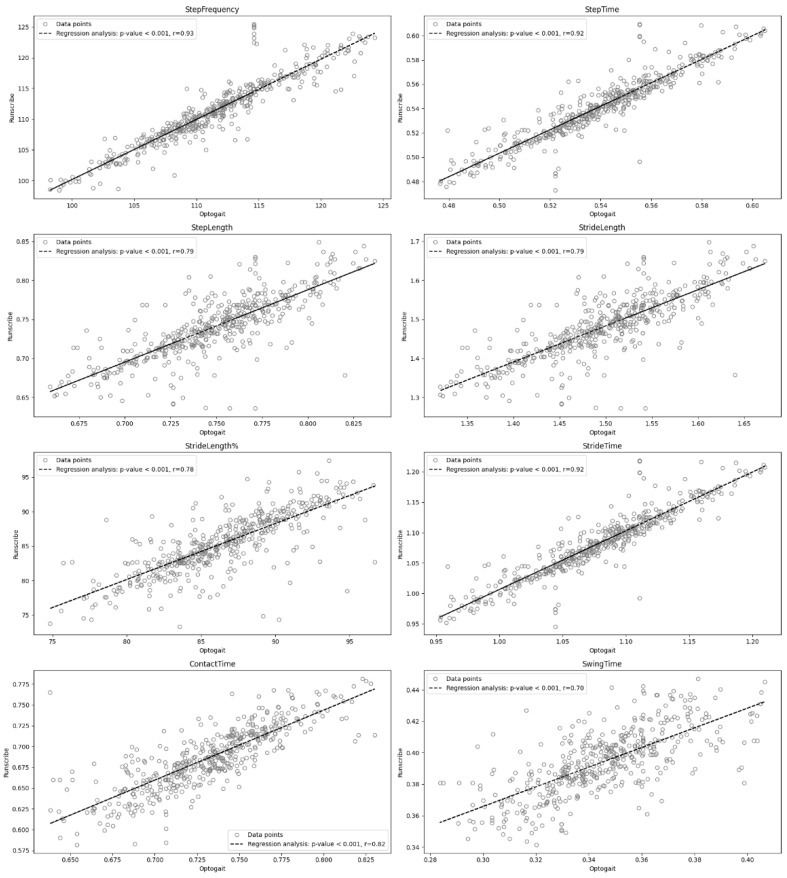
Linear regression model.

**Table 1 sensors-24-07825-t001:** Descriptive characteristics of the participants.

Variable		Mean ± SD
Age (years)		36.32 ± 13.10
Height (cm)		173.47 ± 9.00
Body Mass (kg)		70.14 ± 13.14
Sex	Male Female	318 (69%) 142 (31%)

**Table 2 sensors-24-07825-t002:** Bland–Altman results.

Measures	Mean Diff.	SD of Diff.	LB	UB	% Within Limits of Agreement
SF (steps/min)	0.07	2	−3.85	3.98	94.8
ST (s)	0.00	0.01	−0.02	0.02	94.3
SL (m)	0.01	0.02	−0.04	0.06	92.6
StL (m)	0.02	0.05	−0.08	0.11	92.6
StL% (%)	1.07	2.83	−4.48	6.62	93.5
StT (s)	0.00	0.02	−0.05	0.04	94.3
CT (s)	0.05	0.02	0.00	0.09	94.1
SwT (s)	−0.05	0.02	−0.08	−0.01	93.7

Note: LB and UB represent the limits of agreement of the Bland–Altman analysis.

**Table 3 sensors-24-07825-t003:** Comparative analysis of gait variables during walking from the OptoGait™ and RunScribe™ systems.

Measures	Mean ± SD (OptoGait)	Mean ± SD (RunScribe)	*p*-Value (*t*-Test)	Effect Size (Cohen’s d)	Pearson Corr. (r (*p*-Value))	ICC (2,k)
**SF (steps/min)**	111.30 ± 5.27	111.23 ± 5.53	0.849	0.013	0.93 (<0.001)	0.965 [0.96–0.97]
**ST (s)**	0.54 ± 0.03	0.54 ± 0.03	0.28	−0.071	0.92 (<0.001)	0.957 [0.95–0.96]
**SL (m)**	0.75 ± 0.03	0.74 ± 0.04	0.001	0.22	0.79 (<0.001)	0.868 [0.82–0.90]
**StL (m)**	1.50 ± 0.07	1.48 ± 0.08	0.001	0.218	0.79 (<0.001)	0.868 [0.82–0.90]
**StL% (%)**	86.48 ± 4.16	85.42 ± 4.34	<0.001	0.251	0.78 (<0.001)	0.860 [0.80–0.90]
**StT (s)**	1.08 ± 0.05	1.08 ± 0.05	0.28	−0.071	0.92 (<0.001)	0.957 [0.95–0.96]
**CT (s)**	0.73 ± 0.04	0.69 ± 0.04	<0.001	1.213	0.82 (<0.001)	0.641 [−0.18–0.88]
**SwT (s)**	0.35 ± 0.02	0.39 ± 0.02	<0.001	−2.195	0.70 (<0.001)	0.340 [−0.11–0.70]

## Data Availability

The data presented in this study are available on request from the corresponding author due to ongoing research.

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
