# Peer review of "Concurrent Validity and Relative Reliability of the RunScribe™ System for the Assessment of Spatiotemporal Gait Parameters During Walking"

_sensors, 2024, doi:10.3390/s24237825_

Round 1

Reviewer 1 Report

Comments and Suggestions for Authors

Summary: The authors compared the performance a portable IMU unit (RunScribe) versus a photocell system (OptoGait) in measuring the spatiotemporal gait parameters walking on treadmill at constant speed. The authors aim to demonstrate the RunScribe system can provide the same quality of data as a commonly used OptoGait system. The strength of the study the large, sample size of the study, which give credence to both the positive and negative results of the study. However, there are several design flaws in the study.  The scientific rigor was lacking in the study design.  The study also does not provide sufficient the current state of knowledge to see the value of the RunScribe system.

Additional Comments:

Introduction:

§  Introduction lacking the reason to perform the study, while walking is important, the authors do not present an argument why portable imu I needed.

§  The introduction lacks the current state of knowledge in the field. Depending on how the authors define the term portable, there are numerous sensors and companies already in the market and who are already embedded in clinical trials, sport performance, and methodology to measure gait during real-living environment. It would be beneficial to provide the overview of the current state of knowledge.

o   APDM – Opal system

o   BioSensics – LEGSys system

o   Nike+

o   Reev Sense

o   Gait-up (physiology)

o   etc…

§  It is not clear what the gaps in knowledge in this study because there is a lack of literature review on the process.

§  Similarly, there was already a publication (Garcia-Pinillos et al, 2019, Plos ONE) which showed the agreement between the RunScribe and a high-speed video analysis.  The study was also done on the same treadmill (Woodway PRO XL).  The only difference is now the study was conducted with OptoGait and with larger  sample. That study also showed high correlation for contact time, swing time, step frequency and step length.

§   

Method

o   The study is limited to walking on treadmill and not provide differentiation from the previous study which also validate the RunScribe system.  The only difference is in the increased in sample size.

o   Define null data set – which the flow diagram (Figure 1 ) unclear.  Is the null data referring to data collected with 1 device?  The use of them null data can be confusing to the reader, would missing data be more appropriate use here?

o   It is not clear on the design of the system, the aim of the study is to look at the reliability and validity of RunScribe, it would have been more scientific rigor to compare it to the gold-standard system such as VICON.  By the author explanation, the OptoGait system is not able to capture accurately some of the spatiotemporal parameter such as contact  and swing time.  The study design seems to be a result of convenient rather than based on scientific rigor.

o   The statistical analysis plan was well designed.

o   What is the rationale for using a fix 5 km/h speed vs preferred walking speed?

Results

o   The step frequency is denoted as Steps/min – however, looking at Figure 3 (Bland-Altman plot), the mean is 2.0 – 2.08, is this 2.0 – 2.8 steps/min (this seems low)

o   While stride length (SL has an ICC = .79), this seems low for a technical equivalence given that OptoGait has excellent agreement while OptoGait has an ICC of .991 vs GAITRite (Lee, Myung Mo, et al. 2014). This highlighted the flaw in the design process to compare the device to a non-gold-standard system.

o   Table 4. ICC(3k), suggest using ICC(3,k) for clarity

Conclusion:

o   The conclusion does not seem to support the statement that RunScribe provide valid and reliable spatiotemporal data giving that the correlation for some parameters are really low, particular for parameters in the temporal area (contact time, swing time). For walking, these two parameters are important in analyzing gait disorders. And given the discussion that this might be a downsize of the OptoGait, the authors using this system for comparison is not justifiable.

Author Response

Summary: The authors compared the performance a portable IMU unit (RunScribe) versus a photocell system (OptoGait) in measuring the spatiotemporal gait parameters walking on treadmill at constant speed. The authors aim to demonstrate the RunScribe system can provide the same quality of data as a commonly used OptoGait system. The strength of the study the large, sample size of the study, which give credence to both the positive and negative results of the study. However, there are several design flaws in the study.  The scientific rigor was lacking in the study design.  The study also does not provide sufficient the current state of knowledge to see the value of the RunScribe system.

Thanks for your insightful comments and kind words. Thanks to them our manuscript was substantially improved. We hope that you are satisfied with the changes made.

Additional Comments:

Introduction:

  1. Introduction lacking the reason to perform the study, while walking is important, the authors do not present an argument why portable imu I needed.

We appreciate your observation regarding a more explicit rationale for conducting the study and the importance of portable IMUs. In response, we have revised the introduction section to include a more comprehensive justification.

  1. The introduction lacks the current state of knowledge in the field. Depending on how the authors define the term portable, there are numerous sensors and companies already in the market and who are already embedded in clinical trials, sport performance, and methodology to measure gait during real-living environment. It would be beneficial to provide the overview of the current state of knowledge.
    • APDM – Opal system
    • BioSensics – LEGSys system
    • Nike+
    • Reev Sense
    • Gait-up (physiology)
    • etc…

Thank you for your feedback. In response, we have revised the introduction to include additional information about other sensors and devices currently available on the market, drawing on findings from a recent systematic review and meta-analysis (Kobsar et al, 2020). We believe that the lack of high-quality studies for specific outcomes, as identified in the systematic review—characterized by underpowered and/or unjustified sample sizes (median sample size: 12; range: 2–95)—highlights an important gap in the current literature. In contrast, our study, which employs a robust methodology and a significantly larger sample size (n=460), has the potential to challenge and refine the findings reported in previous research.

  1. It is not clear what the gaps in knowledge in this study because there is a lack of literature review on the process.

Thank you for your comment. We hope that the updated introduction provides greater clarity regarding the rationale of the study.

  1. Similarly, there was already a publication (Garcia-Pinillos et al, 2019, Plos ONE) which showed the agreement between the RunScribe and a high-speed video analysis.  The study was also done on the same treadmill (Woodway PRO XL).  The only difference is now the study was conducted with OptoGait and with larger  sample. That study also showed high correlation for contact time, swing time, step frequency and step length.

Thank you for highlighting this reference. While we acknowledge the similarity in methodology, it is important to note a key difference: the study by García-Pinillos et al. (2019) focused on running, whereas the present study was conducted during walking.

Method

  1. The study is limited to walking on treadmill and not provide differentiation from the previous study which also validate the RunScribe system.  The only difference is in the increased in sample size.

Thank you for your comment. As mentioned previously, the methodology of our study differs from the referenced study, as our focus was specifically on walking, while the prior study validated the RunScribe system during running.

  1. Define null data set – which the flow diagram (Figure 1 ) unclear.  Is the null data referring to data collected with 1 device?  The use of them null data can be confusing to the reader, would missing data be more appropriate use here?

Thank you for your feedback. The term "null data" has been replaced with "missing data". We have updated the flow diagram (Figure 1):

Figure 1. Flowchart of the database.

  1. It is not clear on the design of the system, the aim of the study is to look at the reliability and validity of RunScribe, it would have been more scientific rigor to compare it to the gold-standard system such as VICON.  By the author explanation, the OptoGait system is not able to capture accurately some of the spatiotemporal parameter such as contact  and swing time.  The study design seems to be a result of convenient rather than based on scientific rigor.

Thank you. The study was performed with the resources available in the biomechanical laboratories where the data collection took place. Nonetheless, as described in the study, OptoGait system has been previously validated while walking and running.

  1. The statistical analysis plan was well designed.

Thank you for your kind feedback regarding the statistical analysis. We greatly appreciate your recognition.

  1. What is the rationale for using a fix 5 km/h speed vs preferred walking speed?

Thank you for the question. The decision to use a fixed walking speed was made to facilitate the direct comparison of gait variables across participants under standardized conditions. This approach minimizes the variability introduced by individual differences in preferred walking speed, thereby enhancing the reliability of the comparisons. Moreover, this speed aligns with the recommendations of prior studies, which commonly set a walking speed around 5 km/h as a representative pace for gait analysis (Bohannon, 1997).

Results

  1. The step frequency is denoted as Steps/min – however, looking at Figure 3 (Bland-Altman plot), the mean is 2.0 – 2.08, is this 2.0 – 2.8 steps/min (this seems low)

Thank you for your appreciation. We have updated the figure accordingly.

Figure 3. Bland-Altman plots.

  1. While stride length (SL has an ICC = .79), this seems low for a technical equivalence given that OptoGait has excellent agreement while OptoGait has an ICC of .991 vs GAITRite (Lee, Myung Mo, et al. 2014). This highlighted the flaw in the design process to compare the device to a non-gold-standard system.

Thank you for your comment. In the current study, stride length (SL has an ICC =0,87), indicating good reliability following the Koo and Li guidelines. Regarding the study design, as mentioned before, OptoGait system was previously validated while walking and running. The study was performed with the resources available in biomechanical laboratories, that is the main reason for using OptoGait.

  1. Table 4. ICC(3k), suggest using ICC(3,k) for clarity.

Thank you for your appreciation in Table 4. We have updated the table using the suggested format.

Measures

Mean ± SD (OptoGait)

Mean ± SD (RunScribe)

p-value (t-test)

Effect Size (Cohen's d)

Pearson corr. (r (p-value))

ICC (2,k)

SF (steps/min)

111.30 ± 5.27

111.23 ± 5.53

0.849

0.013

0.93 (<0.001)

0.965 [0.96-0.97]

ST (s)

0.54 ± 0.03

0.54 ± 0.03

0.28

-0.071

0.92 (<0.001)

0.957 [0.95-0.96]

SL (m)

0.75 ± 0.03

0.74 ± 0.04

0.001

0.22

0.79 (<0.001)

0.868 [0.82- 0.90]

StL (m)

1.50 ± 0.07

1.48 ± 0.08

0.001

0.218

0.79 (<0.001)

0.868 [0.82-0.90]

StL% (%)

86.48 ± 4.16

85.42 ± 4.34

<0.001

0.251

0.78 (<0.001)

0.860 [0.80- 0.90]

StT (s)

1.08 ± 0.05

1.08 ± 0.05

0.28

-0.071

0.92 (<0.001)

0.957 [0.95-0.96]

CT (s)

0.73 ± 0.04

0.69 ± 0.04

<0.001

1.213

0.82 (<0.001)

0.641 [-0.18- 0.88]

SwT (s)

0.35 ± 0.02

0.39 ± 0.02

<0.001

-2.195

0.70 (<0.001)

0.340 [-0.11-0.70]

Table 3. Comparative analysis of gait variables during walking from the OptoGait™ and RunScribe™ systems.

Conclusion:

  1. The conclusion does not seem to support the statement that RunScribe provide valid and reliable spatiotemporal data giving that the correlation for some parameters are really low, particular for parameters in the temporal area (contact time, swing time). For walking, these two parameters are important in analyzing gait disorders. And given the discussion that this might be a downsize of the OptoGait, the authors using this system for comparison is not justifiable.

Thank you for your thoughtful feedback regarding the conclusion. We have carefully considered your comments and revised the conclusion to address your concerns:

“This study provides evidence for the validity and relative reliability of the RunScribe™ system for assessing several spatiotemporal parameters during walking. However, the findings indicate variability in the accuracy of temporal parameters, particularly contact and swing times. This limitation underscores the need for caution when interpreting these specific metrics and suggests the importance of further validation studies using gold standard systems. The development of accurate wearable devices opens new possibilities for widespread use in healthcare contexts, potentially reducing the barriers to continuous gait monitoring and early detection of mobility issues. As a future line, it would be interesting to evaluate its accuracy during overground walking.”

Reviewer 2 Report

Comments and Suggestions for Authors

This paper makes a significant contribution to the field of gait analysis by validating the RunScribe system for walking, bridging an important gap in the literature (test had been performed for running only). The study’s large sample size of 460 participants and its rigorous methodology, including Bland-Altman and intraclass correlation analyses, provide robust evidence supporting the system's accuracy and reliability for measuring spatiotemporal gait parameters. Furthermore, this research highlights the device's applicability for continuous gait monitoring and early detection of gait disorders, making it a valuable tool for both clinical and everyday applications.

The authors cite 4 of their own works out of the total 29 references in the document. While, in my opinion, these self-citations are appropriate and relevant to the content, this detail caught my attention as it highlights their significant contribution to the field being discussed.

I just have two significant comments I considered should be addressed by authors.

1.     Given that walking and running involve distinct biomechanical patterns, assessing the RunScribe system under walking conditions could provide a more comprehensive understanding of its applicability in everyday scenarios and potential clinical contexts. Could you elaborate on the necessity of evaluating the system during walking if its functionality has already been validated for running? What specific insights or benefits could be gained from examining its performance under walking conditions?

2.     The present study uses a photoelectric cell system as the gold standard, while other studies employed high-speed video analysis, introducing a methodological difference that could lead to variations in measured spatiotemporal parameters. These differences, along with potential variability in sensor placement, environmental conditions, or reference system calibration, could introduce systematic biases or random errors, impacting the performance and comparability of the RunScribe system across applications. You could address how these factors affect comparisons between studies and suggest future research to standardize reference systems for gait analysis, improving cross-study consistency and reliability.

Author Response

This paper makes a significant contribution to the field of gait analysis by validating the RunScribe system for walking, bridging an important gap in the literature (test had been performed for running only). The study’s large sample size of 460 participants and its rigorous methodology, including Bland-Altman and intraclass correlation analyses, provide robust evidence supporting the system's accuracy and reliability for measuring spatiotemporal gait parameters. Furthermore, this research highlights the device's applicability for continuous gait monitoring and early detection of gait disorders, making it a valuable tool for both clinical and everyday applications.

The authors cite 4 of their own works out of the total 29 references in the document. While, in my opinion, these self-citations are appropriate and relevant to the content, this detail caught my attention as it highlights their significant contribution to the field being discussed.

I just have two significant comments I considered should be addressed by authors.

  1. Given that walking and running involve distinct biomechanical patterns, assessing the RunScribe system under walking conditions could provide a more comprehensive understanding of its applicability in everyday scenarios and potential clinical contexts. Could you elaborate on the necessity of evaluating the system during walking if its functionality has already been validated for running? What specific insights or benefits could be gained from examining its performance under walking conditions?

We sincerely appreciate your insightful comments. As you mentioned, the biomechanics of walking and running are fundamentally different, and for this reason, it cannot be assumed that a system validated for running would automatically be reliable for walking. This means that certain parameters, such as stance time (longer in walking) or swing time (defined as “flight time” for running, and longer in running) may be more prominent or differently expressed during walking. Additionally, walking is a key activity for a wide range of populations, including older adults, individuals recovering from injury, or those with chronic conditions. Validating the system for walking could ensure its applicability for these groups, expanding its clinical applicability.

  1. The present study uses a photoelectric cell system as the gold standard, while other studies employed high-speed video analysis, introducing a methodological difference that could lead to variations in measured spatiotemporal parameters. These differences, along with potential variability in sensor placement, environmental conditions, or reference system calibration, could introduce systematic biases or random errors, impacting the performance and comparability of the RunScribe system across applications. You could address how these factors affect comparisons between studies and suggest future research to standardize reference systems for gait analysis, improving cross-study consistency and reliability.

Thank you foy your comments. Effectively, the study was performed in comparison with OptoGait. The study was performed with the resources available in the biomechanical laboratories where the data collection took place. Nonetheless, as described in the study, OptoGait system has been previously validated while walking and running. The possible difference between systems may be explained by Lienhard et al.: “the LEDs of the OptoGait™ system are positioned 3 mm above ground, causing heel contact to be detected earlier and toe lift-off to be detected later in the gait cycle. This leads to the recording of longer contact times and shorter swing times”. Regarding the sensor placement, a previous study analysed the influence of the RunScribe position while running, indicating that “the lace shoe placement showed smaller systematic bias, random errors, and narrower limits of agreement for contact time, flight time, and step length.” These indications were followed for the data collection and described in the section 2.3.

Reviewer 3 Report

Comments and Suggestions for Authors

INTRODUCTION

- The first chapter effectively emphasizes the importance of ambulation and gait analysis. It would be helpful, perhaps, to have a brief statement about the economic impact of gait abnormalities to also illustrate their importance.

- The shift from visual inspection to the use of quantitative analysis in the second paragraph could be smoother. Consider expanding on why visual inspection alone doesn't work.

- The description of the OptoGait™ system is very detailed but could be more condensed in the introduction. Focus on its general purpose and popularity, and leave the technical details for the methods section.

- The paragraph on IMUs and the RunScribe™ system is well written and clearly identifies the research gap.

- The final paragraph effectively states the research aim. Consider adding a brief statement about the potential implications or applications of this research.

MATERIALS AND METHODS

- It would be enlightening to have more information on participant recruitment and from what population.

- Whether randomization was part of the study design must be indicated.

- More details about the two biomechanical laboratories and the steps taken to maintain consistency between these locations would be useful.

- While a 10-minute acclimatization phase is indicated, it would be good to have the rationale for this duration supported by previous studies.

- More detail might be specified in relation to the types of shoes allowed and how these can affect measurements.

- It would be good to have more detail on how the raw data from both systems was processed and matched.

RESULTS

- In section 3.1, add a short note on what is meant by a "small" systematic bias and random error to aid readers not familiar with Bland-Altman analysis.

- In section 3.2, it is necessary to proceed the sentence "T-test demonstrated no significant differences between those systems in SF (p=0.822), ST (p=0.28) and StT (p=0.28)" with the discussion concerning the variables which showed significant differences.

- The large effect sizes for CT and SwT certainly deserve more attention, perhaps with some brief comment as to the reasons for these differences.

DISCUSSION

- It would be nice to have more explicit statements regarding the clinical implications of these findings.

- Although the limitations are well-discussed, the authors could offer more concrete ideas for future research.

- Authors should discuss if findings are generalizable to other populations or generalizable to different walking environments.

Author Response

INTRODUCTION

- The first chapter effectively emphasizes the importance of ambulation and gait analysis. It would be helpful, perhaps, to have a brief statement about the economic impact of gait abnormalities to also illustrate their importance.

Thank you for your observations. We agree on the importance of highlighting the economic impact of gait abnormalities. Consequently, we have added information to emphasize this aspect:

“In addition, clinical gait analysis plays a critical role in clinical decision making, particularly in the context of surgery. Research has demonstrated that incorporating gait analysis can refine proposed surgical procedures, lead to changes in final decisions, and ultimately reduce the costs associated with surgery, highlighting its economic impact [8-10].”

- The shift from visual inspection to the use of quantitative analysis in the second paragraph could be smoother. Consider expanding on why visual inspection alone doesn't work.

Thanks. We have updated the second paragraph explaining briefly why the visual inspection alone doesn’t work on biomechanical analysis:

“However, this qualitative assessment is often subjective, and therefore unreliable and in-accurate for biomechanical analysis. The literature suggests that a quantitative analysis …”

- The description of the OptoGait™ system is very detailed but could be more condensed in the introduction. Focus on its general purpose and popularity, and leave the technical details for the methods section.

Thank you for the comment. We have summarized the description of the OptoGait™ system, removing the technical details (explained in the methods section):

“In particular, the OptoGait™ system has gained popularity among sports scientists and clinicians for its ability to capture and analyze spatiotemporal gait parameters.”

- The paragraph on IMUs and the RunScribe™ system is well written and clearly identifies the research gap.

Thank you for your positive feedback. We greatly appreciate your recognition.

- The final paragraph effectively states the research aim. Consider adding a brief statement about the potential implications or applications of this research.

Thank you. The following text has been added at the end of the introduction:

“This study will provide valuable insights into the real-time, real-world assessment of gait dysfunctions, demonstrating the potential of low-cost wearable devices as reliable and valid tools for gait analysis.”

MATERIALS AND METHODS

- It would be enlightening to have more information on participant recruitment and from what population.

Thank you. We have added more information on participant recruitment:

“A sample of young and middle-aged adults between the ages of 20 and 50 with a low potential risk for significant gait abnormalities was recruited. Individuals who voluntarily agreed to participate in the biomechanical gait study and met the inclusion criteria were ultimately evaluated.”

- Whether randomization was part of the study design must be indicated.

Thank you for your consideration. All the participants were measured with the two systems and randomization was not part of the study design. Further considerations regarding the recruitment phase have been added in the material and methods section for the sake of clarity.

- More details about the two biomechanical laboratories and the steps taken to maintain consistency between these locations would be useful.

Thank you: The following information has been added to the manuscript:

“Two experienced researchers with expertise in the use of RunScribe and OptoGait collaborated in the design and implementation of the experimental protocol. Data collection was later extended to a second laboratory. Both researchers strictly followed the calibration and setup protocols provided by the manufacturers to ensure consistency between the two biomechanical laboratories.”

- While a 10-minute acclimatization phase is indicated, it would be good to have the rationale for this duration supported by previous studies.

Thank you. Following your consideration it has been made clear that the 10-minute acclimatization was supported by the following study: Van de Putte, M.; Hagemeister, N.; St-Onge, N.; Parent, G.; de Guise, J.A. Habituation to Treadmill Walking. Biomed Mater Eng 2006, 16, 43–52.

- More detail might be specified in relation to the types of shoes allowed and how these can affect measurements.

Thank you for the observations. The following paragraph has been added to the text:

Participants were advised to wear their usual footwear to enhance the ecological validity of the study and minimize potential gait alterations that could result from using shoes different from their normal choice for a long walk. Therefore, minimalist shoes were not permitted, and participants were encouraged to wear athletic or running shoes instead.

- It would be good to have more detail on how the raw data from both systems was processed and matched.

Thank you, the following explanation has been added in the corresponding section:

"A total of 90 seconds of data were recorded simultaneously using both instruments. For analysis, the first and last 15 seconds of each capture were excluded, thus focusing on the central minute. The average values for each subject were included in the comparison between devices, with the data matched using the timestamp and subject code.”

RESULTS

- In section 3.1, add a short note on what is meant by a "small" systematic bias and random error to aid readers not familiar with Bland-Altman analysis.

Thanks. We agree, that clarifying what constitutes a 'small' systematic bias and random error will enhance the accessibility of our Bland-Altman analysis for readers unfamiliar with the method. We included a brief explanation and referenced a relevant guideline for context.

- In section 3.2, it is necessary to proceed the sentence "T-test demonstrated no significant differences between those systems in SF (p=0.822), ST (p=0.28) and StT (p=0.28)" with the discussion concerning the variables which showed significant differences.

- The large effect sizes for CT and SwT certainly deserve more attention, perhaps with some brief comment as to the reasons for these differences.

Thank you. Following your recommendation we included further considerations in section 3.2 to discuss more about the significant variables and the effect size.

DISCUSSION

- It would be nice to have more explicit statements regarding the clinical implications of these findings.

Thank you for your valuable feedback. We have considered your comments and updated the discussion:

“From a practical point of view, and given the widespread use of both systems, it is essential for scientists and clinicians to recognize that both devices have demonstrated adequate reliability for walking assessment and, therefore spatiotemporal parameters reported by these devices can be compared over time (when using the same device). However, it is equally important for users to understand the limitations of directly comparing data between these two devices, particularly in the assessment of contact and swing times. The results of the present study hold significant implications for clinicians and researchers utilizing IMUs to assess and investigate gait abnormalities. Future research should explore the applicability of these results to other populations, such as individuals with mobility impairments or different age groups and investigate their appropriateness in over-ground conditions.”

- Although the limitations are well-discussed, the authors could offer more concrete ideas for future research.

 Thanks. As mentioned in the previous comment, new information has been added to include future research areas of interest.

- Authors should discuss if findings are generalizable to other populations or generalizable to different walking environments.

Thank you. The following text has been added to the corresponding section to discuss the generalizability of the findings:

“The large sample size included in this study makes its findings of great value and enhances its validity among larger populations of young adults.”

Round 2

Reviewer 1 Report

Comments and Suggestions for Authors

Thank you the author(s) for their details responses and I agree that their responses are reasonable based on the additional context of the study.  

minor comment:

Figure 4 - linear regression.  The author suggest removing the equation for the regression line since this equation does not provide additional usage beyond  the Pearson correlation constant.  The equation is only useful if you have an input and you want to calculate the output.  would be more helpful to provide the r and p value for each comparison. 

Author Response

Comments and Suggestions for Authors

Thank you the author(s) for their details responses and I agree that their responses are reasonable based on the additional context of the study.  

minor comment:

  1. Figure 4 - linear regression.  The author suggest removing the equation for the regression line since this equation does not provide additional usage beyond  the Pearson correlation constant.  The equation is only useful if you have an input and you want to calculate the output.  would be more helpful to provide the r and p value for each comparison. 

Thank you for your valuable feedback regarding Figure 4. We have carefully reviewed your suggestion and have made the requested modification. Specifically, we have removed the equation for the regression line, and we have included the r and p-values for each comparison to provide more relevant and informative insights.

Figure 4. Linear regression model.
